# Adverse Childhood Experiences and Chronic Diseases: Identifying a Cut-Point for ACE Scores

**DOI:** 10.3390/ijerph20021651

**Published:** 2023-01-16

**Authors:** Fahad M. Alhowaymel, Karen A. Kalmakis, Lisa M. Chiodo, Nicole M. Kent, Maha Almuneef

**Affiliations:** 1Department of Nursing, College of Applied Medical Sciences, Shaqra University, Shaqra 11911, Saudi Arabia; 2Elaine Marieb College of Nursing, University of Massachusetts Amherst, Amherst, MA 01003, USA; 3College of Medicine, King Saud bin Abdulaziz University for Health Sciences, Riyadh 14611, Saudi Arabia; 4King Abdullah International Medical Research Center, Riyadh 11481, Saudi Arabia

**Keywords:** Adverse Childhood Experiences, childhood trauma, cut-point, chronic disease

## Abstract

Adverse Childhood Experiences (ACEs) contribute to many negative physiological, psychological, and behavioral health consequences. However, a cut-point for adverse childhood experience (ACE) scores, as it pertains to health outcomes, has not been clearly identified. This ambiguity has led to the use of different cut-points to define high scores. The aim of this study is to clarify a cut-point at which ACEs are significantly associated with negative chronic health outcomes. To accomplish this aim, a secondary analysis using data from a cross-sectional study was conducted. The Adverse Childhood Experiences-International Questionnaire (ACE-IQ) was used for data collection. Descriptive statistics, nonparametric regression, and logistic regression analyses were performed on a sample of 10,047 adults. Data from demographic and self-report health measures were included. The results showed that a cut-point of four or more ACEs was significantly associated with increased rates of chronic disease. Participants with at least one chronic disease were almost 3 times more likely (OR = 2.8) to be in the high ACE group. A standardized cut-point for ACE scores will assist in future research examining the impact of high ACEs across cultures to study the effect of childhood experiences on health.

## 1. Introduction

“Adverse Childhood Experiences” (ACEs) refers to a wide range of adversities that cause stress in children [1]. ACEs include childhood abuse (psychological, emotional, physical, and sexual abuse) and household dysfunction (substance misuse or mental illness affecting members of the family, domestic violence, and criminal behavior) [2,3]. Other types of ACEs include physical and emotional neglect [3]; witnessing verbal, physical, and/or sexual mistreatment of other family members [4]; community violence [5]; and having an incarcerated household member [6]. ACEs are characterized as events that occur before the age of 18, repeat over time, and vary in intensity [6,7]. ACEs contribute to many negative health consequences including, but not limited to, physiological, psychological, and behavioral health [8]. However, evidence for the cut-point at which ACEs raise an individual’s risk for chronic disease to a significant level has not been established.

There are multiple ACE questionnaires that consist of anywhere from 10 [9] to 31 items [1] used to query about abuse, neglect, household dysfunction, and community violence. For this study, the researchers wanted to find the threshold number at which ACEs are associated with health outcomes, regardless of the total ACE items. We hypothesize that there is a cut-point at which the association between ACEs and chronic disease is different above and below that cut-point. Such an ACE threshold will benefit researchers by providing an evidence-based cut-point to indicate high ACE scores.

Researchers report a significant relationship between ACEs, neurodevelopment, and long-term health [10,11]. The chronic nature of childhood adversities may lead to allostatic load and resulting cardiovascular, endocrine, and immune system dysfunction [12,13,14]. Indeed, ACE research over the past few decades provides evidence for a relationship between ACEs and diabetes [15], depression [16], anxiety [17], mental disorders [18], and obesity [19].

Since the original ACE study conducted in the US [9], nursing, public health, medicine, social service, and criminal justice researchers have studied the relationship between ACEs and health [15], and have expanded the research globally [20]. However, a consistent cut-point for the number of ACEs sufficient to show a significant relationship to negative health outcomes remains unclear. Although many studies have used a cut-point of 4 ACEs [21], researchers have used various ACE score cut-points from 2 ACEs [22] to 3 ACEs [23] to 5 ACEs [24], as well as multiple cut-points within the same study [25] to indicate adversity sufficient to affect health. This ambiguity leads to confusion and weakness in ACEs measurements and increases the need for establishing a cut-point score for ACEs [26]. Thus, the aim of this study is to identify a cut-point for “high”-level ACEs— the point at which ACEs are significantly associated with negative chronic health outcomes.

## 2. Materials and Methods

### 2.1. Study Design

This study is a secondary analysis of a Saudi national dataset. The original study used a cross-sectional design to collect data from people living in Saudi Arabia to examine ACEs and specific health outcomes.

### 2.2. Participants

The data analyzed for this study was obtained from the National Family Safety Program (NFSP) in Saudi Arabia. In 2013, the NFSP staff members collected information about ACEs as part of an international collaboration with various nations and the World Health Organization (WHO). ACEs and chronic disease information were collected as part of the NFSP from males and females aged 18 years in several nations globally, including Saudi Arabia.

### 2.3. Data Collection

The data was collected from the 13 provinces of Saudi Arabia; 182 different locations in those provinces were randomly selected on the basis of the geographical classification in the country. The purpose was to select and survey at least one large and one small city from each province to have a representative sample of the population in the kingdom. The data collection team prepared booths at public locations such as shopping centers, public parks, and primary care clinics to recruit participants. Eligible participants who met the inclusion criteria and agreed to participate signed a written consent form. The questionnaire was distributed, and the participants were provided a private place to complete them and drop them in a secure box [27,28]. The original dataset had a total of 10,156 participants; 109 of these were excluded from the current study because they were not Saudi. The total study sample for these analyses was 10,047. In sum, the sample is representative of the Saudi population, as all the 13 provinces of Saudi Arabia were surveyed taking into consideration geographical, cultural, and norm variations.

### 2.4. Measurement

An adapted version of the Adverse Childhood Experiences-International Questionnaire (ACE-IQ) was used for data collection. The ACE-IQ was developed by a team of experts from the WHO and the Centers for Disease Control and Prevention (CDC) and international experts in the field [1]. This self-report instrument is designed for subjects 18 years old or older. ACE-IQ includes 29 items representing different types of ACE. Examples of ACE items include questions regarding family environment; emotional, physical or sexual abuse; peer violence; witnessing community violence; and exposure to war or collective violence. As previously mentioned, this study is a secondary analysis of prior research [5]. In the prior research, the ACE-IQ category “collective violence” was removed as it was inconsistent with the culture of Saudi Arabia [5]. A prior study conducted in South Korea also excluded collective violence for similar reasons [29]. Thus, the remaining 25 items within 12 categories were included in this analysis. The adapted ACE-IQ score ranged from 0 to 25.

The ACE-IQ has been found to have adequate reliability and validity when used in several countries including China (r = 0.29; [30]) and Nigeria (r = 0.72; [31]). In Saudi Arabia, the ACE-IQ was first piloted with 200 participants to assess its cultural and social adaptability and accessibility [5]. The test was translated into Arabic and then back-translated to English, with some modifications for cultural adaptability, and validated by Almuneef and colleagues (2014) in their research in Saudi Arabia [5].

The variables that were used and analyzed in this study include demographic covariates (gender, age, education, occupation, marital status, and geographical setting), ACE-IQ total score, and total number of chronic diseases (diabetes, hypertension, coronary heart disease, chronic respiratory disease, liver disease, obesity, and depression).

### 2.5. Ethical Consideration

The design of the original data collection was approved by a Human Subjects Review Board in Saudi Arabia. The researchers of the current study obtained the dataset from the primary investigator of the original study. All the responses were anonymous and used for research purposes only. Moreover, the results did not include the respondents’ personal health information.

### 2.6. Data Analysis

Data analysis was completed using the Statistical Package for Social Sciences (SPSS) version 28. Descriptive statistics for demographic information, ACEs, and chronic disease prevalence were calculated to determine frequencies and percentages. Nonparametric regression analysis was performed to identify the presence of a cut-point for ACE and the rate of chronic disease. Finally, a logistic regression was performed to examine the relationship between the two ACE groups (low/high) and the likelihood of chronic disease. In the logistic regression, several demographic variables were included as covariates (gender, age, marital status, education, and employment). Since employment was a 3-group nominal variable (unemployed/retired, employed, student), two dummy coded variables were created (employed vs other and student vs other).

## 3. Results

### 3.1. Sample Characteristics

A majority of the 10,047 respondents (64%) were between 18 and 37 years old (mean = 34.3; SD = 11.3). The sample was 52% male, and most of the participants were from urban settings (86.8%). Just over half of the sample (58.6%) had a high school education or less, about half (51.6%) were employed, and 58.7% were married (see Table 1).

### 3.2. Prevalence of ACEs and Chronic Diseases

Among the respondents, the average number of ACEs was 5.8 (SD = 5.0, range = 0–25), and 87.6% of the respondents reported having at least one ACE. The top 10% of the sample had 13 or more ACEs, the top 5% had 15 or more ACEs, and two respondents were positive for all 25 ACEs. Additional information about frequency of ACEs in this sample is provided in Table 2. Almost 40% of the sample reported being diagnosed with at least one chronic disease; approximately 20% had been diagnosed with two or more. Hypertension and diabetes were the most frequently reported chronic diseases in this cohort (20% and 17.1%, respectively).

### 3.3. Identification of a Cut-Point for a High ACE Level

Nonparametric regression analysis was performed to identify the presence of a cut-point between the total number of ACEs and the frequency of chronic diseases (see Figure 1). Visual inspection of the nonparametric regression line shows no association between ACEs and chronic disease until there is a total of 4 ACEs or more. If a person has 4 ACEs or more, the regression line begins to increase in a linear fashion with no other instance of flattening. This pattern supports the cut-point of 4 in the relationship between ACEs and the frequency of chronic disease in this sample.

### 3.4. Relationship between ACEs and Chronic Disease

A logistic regression was performed to examine the relationship between the ACE group and the likelihood that participants had a chronic disease. The ACE group was constructed to <4 ACEs and ≥4 ACEs. Analyses controlled for several demographic characteristics including age, gender, education, employment, geographic setting, and marital status. The logistic regression model was statistically significant, χ^2^(8) = 1098.6, *p* < 0.001, and the model accounted for 14.6% of the variance in the frequency of chronic disease. Overall, the participants with at least one chronic disease were almost 3 times more likely (OR = 2.8) to be in the high ACE group. Additionally, both age and gender were significant predictors of chronic disease frequency. With every increase in age group, a participant was 1.1 times more likely to have a chronic disease, while men were 31.7% less likely to have a chronic disease compared to women in this sample (see Table 3).

## 4. Discussion

The lack of a consistent cut-point for ACE scores has led to the use of different cut-points across studies. We successfully determined the cut-point for ACEs in relation to the frequency of chronic disease. The results support that an ACE score of four or more is significantly associated with higher probability of chronic disease. We further propose that this cut-point can be used regardless of the version of ACE measure used. Thus, ≥ 4 ACEs is a risk level for negative health outcomes.

Uniquely in this study, we examined the presence of a cut-point for ACEs in relation to specific chronic diseases. Our contribution to the science of ACEs and health is the identification of a point at which ACE scores are significantly associated with negative chronic health outcomes. To date, researchers have used varying cutoff scores for ACEs without justification. Through this study, we have identified a cut-point of four ACEs as the point at which there was a significant association between ACEs and chronic health conditions. The association between ACEs and chronic diseases emerged at four ACEs and continued increasing in a linear fashion as the ACEs increased beyond four. This result supports the use of four ACEs as a cutoff score to delineate high ACE scores when examining ACEs and chronic disease.

The average number of ACEs (mean = 5.8) in the study sample is consistent with prior research in other developing countries such as in China [32], the Philippines [33], and Brazil [34]. Hypertension and diabetes were the most frequently reported chronic diseases among the participants in this study. Obesity, diabetes, and hypertension have been identified as the top five health risk factors in Saudi Arabia [35]. Approximately one in three Saudi adults suffer from multiple chronic conditions [36], although the burden of disease decreased between 1990 and 2017 [35]. Indeed, chronic disease is among the major public health issues in Saudi Arabia [37]. Underuse of medical services during young adulthood may contribute to the high prevalence and burden of risk factors, including chronic diseases, among the Saudi population [35,38].

The results of this study support the use of an ACE score of four or more when testing the relationship between ACEs and chronic diseases. Indeed, study participants with chronic diseases were nearly three times more likely to be in the **>**4 ACE group. This result is consistent with studies worldwide. For example, among Iraqi participants there was a 98% increase in chronic physical diseases among those who reported household dysfunction and an 81% increase among those who reported abuse [39]. Among American participants, ACEs were predictors of heart disease, stroke, and chronic obstructive pulmonary disease [40]. ACEs are believed to result in lifelong biological and psychological changes that lead to negative health outcomes [40,41,42,43], providing a possible explanation for the significant relationship between ACEs and chronic diseases.

### Limitations and Strengths

There are limitations to the current study. First, this is a secondary analysis of existing data; therefore, the study is limited to the health outcomes, measures, and populations of the original project [44]. Second, the ACE information was collected retrospectively; this method could have resulted in recall bias and incorrect reporting of ACEs, and this is a weakness of self-report measures [45]. Additionally, it should be noted that Saudi Arabia, like any other country, has unique social, cultural, and religious norms in which children live, learn, and grow. This study adds to the knowledge of ACEs and health globally. The use of a large and representative sample size is another strength of the study.

## 5. Conclusions

An established ACE cut-point, considered sufficient to affect health, was identified in this study. The identified ACE cut-point of four or more will provide a guide for future ACE research. This cut-point for high ACEs should be tested in future studies with different populations and health conditions. Evidence for the prevalence of ACEs and chronic diseases in Saudi Arabia has also been provided. Healthcare systems should focus on developing and introducing strategies and programs for early detection and prevention of ACEs to enhance quality of life for people globally. Screening for ACEs in health care should be the first step to identify people at risk for the health consequences of ACEs, followed by chronic disease prevention programs to mitigate the burden of chronic disease in Saudi Arabia.

## Figures and Tables

**Figure 1 ijerph-20-01651-f001:**
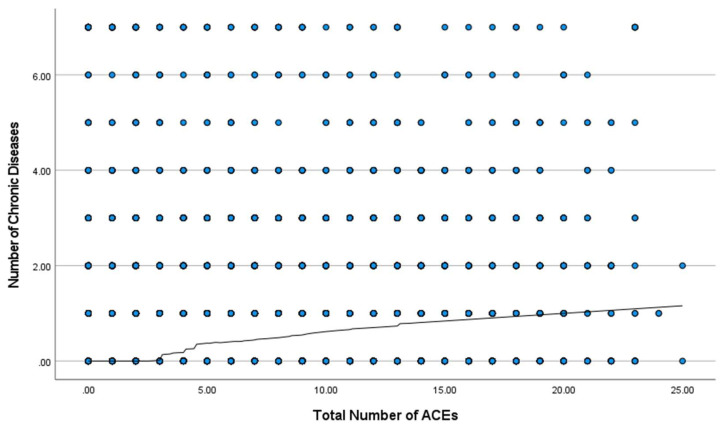
ACEs and chronic disease frequency.

**Table 1 ijerph-20-01651-t001:** Demographic information (N = 10,047).

Variables	N	% *
Age (Mean = 34.3)	10,029	
18–27 years old		34.0
28–37 years old		30.0
38–47 years old		20.0
48–57 years old		13.8
58 years old or older		2.2
Gender	10,028	
Female		47.6
Male		52.4
Geographical setting	10,047	
Urban		86.8
Non-urban		13.2
Education	9938	
High school or below		58.6
College or above		41.4
Occupation	9808	
Unemployed		28.8
Employed		51.6
Students		16.3
Retired		3.4
Marital status	9942	
Married		58.7
Not married		41.3

* Total equals 100.1% due to rounding.

**Table 2 ijerph-20-01651-t002:** Prevalence of ACEs and CD (N = 10,047).

Variables	%	Mean	Std. Dev.
ACEs total score (25 items)		5.77	4.97
0 ACEs	12.4		
1 ACE	10.0		
2 ACEs	10.8		
3 ACEs	8.9		
4 + ACEs	57.9		
Chronic diseases total number (7 diseases)		0.77	1.29
No chronic diseases	60.7		
1 chronic disease	19.7		
2 chronic diseases	11.1		
3 chronic diseases	4.4		
4 + chronic diseases	4.1		
Chronic diseases			
Diabetes	17.1		
Hypertension	20.0		
Coronary heart disease	5.9		
Chronic respiratory disease	12.3		
Liver disease	5.2		
Obesity	4.2		
Depression	12.3		

**Table 3 ijerph-20-01651-t003:** Relationship between ACEs and chronic disease.

Variable	B	S.E.	*p*	Exp(B)
Geographical setting	0.05	0.066	0.427	1.054
Gender	−0.38	0.051	<0.001	0.683
Age	0.05	0.002	<0.001	1.047
Marital status	−0.08	0.054	0.120	0.919
Education	−0.09	0.048	0.071	0.918
Employed vs other	−0.07	0.058	0.199	0.928
Student vs other	0.04	0.083	0.644	1.039
ACEs group (< 4, ≥ 4)	1.04	0.047	<0.001	2.832

## Data Availability

The data presented in this study are available on request from the corresponding author. The data are not publicly available due to privacy of the participants.

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
