# Peer review of "Adverse Childhood Experiences and Chronic Diseases: Identifying a Cut-Point for ACE Scores"

_ijerph, 2023, doi:10.3390/ijerph20021651_

Round 1
Reviewer 1 Report
Review of “Adverse childhood experiences and chronic diseases: Identifying a cut-point for ACE scores”
This manuscript uses a large sample to identify a clinical cutpoint for adverse childhood adversities that marks heightened risk for chronic disease. Strengths of the paper include a large sample size (over 10,000) and the clinical relevance of providing a guide for high trauma dosage. The paper is also generally well written. Unfortunately, there are some significant statistical issues. Thus, while the paper has potential, it does need significant revision.
Below are some comments to help the authors further strengthen the manuscript.
1) Some acknowledgement of the many ways that ACEs have been measured is warranted. The list of items they mention in the opening paragraph are not definitive and Felitti and colleagues themselves (the original ACEs research team) have used varying item sets. There are also other proposed item sets by Finkelhor and colleagues and others. This sample uses the ACE-IQ version developed by the WHO, another alternative. This variation the ability to determine a specific cutoff now and needs to be acknowledged.
2) The sentence “Researchers have used varying ACE scores from 2 or 52 more, to 5 or more to indicate adversity sufficient to affect health.” needs citations. Most of the literature I have seen focuses on the cutpoint of 4 found by the authors.
3) The representative sample is a strength of the paper and this should be further emphasized.
4) As far as can be determined from the presentation of the results, the authors only examined one cutpoint, >4. This is unacceptable and, frankly, very surprising to see in a paper which has at its main goal the establishment of a cutpoint.
The way that cutpoints are determined is through sensitivity and specificity analyses re the relative benefits of each cutpoint (1, 2, 3, 4, 5, etc) are evaluated statistically and the cutpoint chosen which best optimizes both specificity and sensitivity. The authors need to re-do their analyses to present sensitivity and specificity data.
5) The discussion will need revising depending on how the new analyses turn out.
Author Response
Dear Reviewer,
Thank you for the time taking in reviewing our manuscript, and for the valuable comments provided to us. We have addressed the comments and submitted a revised manuscript via portal. The revisions are marked up using “Track Changes”. We believe that the comments have substantially enhanced our manuscript. In the attached file, please find our response to your specific comments.
Thank you again.

Reviewer 2 Report
The study examined the relationship between Adverse Childhood Experiences (ACE) and chronic diseases, in order to identify the cut-point - as an issue that has not yet been examined in depth.
There is no doubt that this is a very important study. Seeking to shed light for research and practice with vulnerable children, about the devastating health outcomes of situations of vulnerability, trauma and neglect among children.
However, there are several revisions that are required for this manuscript to make it suitable for publication.
Introduction
• Please present results of previous studies more actively. The findings as they are presented in the introduction, are presented passively.
• Please present in detail the various studies that suggested a cut-point.
• Is there no significance to the context in which the research was conducted? What does it mean that the study in question was conducted in Saudi Arabia? This should be addressed.
Methodology
• The research question and the hypotheses of the research must be clearly presented. • It should be stated in more detail how the procedure of adapting the research tool to the Arab-Saudi context was done.
Discussion
• Please compare the research results with studies in the field, more actively
• Please try to sharpen the understanding of the research results in the context of Saudi Arabia and then its relevance to the wider contexts.
• Research limitations: It is not clear why secondary analysis of data is a research limitation? Unless the writers indicated what they would have done differently.
• Explain, in what way the data collection would be conceptualized - if the researchers were the ones who would manage it?
• Why is retrospective data collection a limitation? explained Again, what would you do differently?
Author Response

(The authors gave the same response as above.)

Round 2
Reviewer 2 Report
Thank you very much, you did a nice job correcting and improving the manuscript according to the comments.